# Enriched Pea Protein Texturing: Physicochemical Characteristics and Application as a Substitute for Meat in Hamburgers

**DOI:** 10.3390/foods12061303

**Published:** 2023-03-18

**Authors:** Irene Peñaranda, María Dolores Garrido, Purificación García-Segovia, Javier Martínez-Monzó, Marta Igual

**Affiliations:** 1Department of Food Science and Technology, Veterinary Faculty, University of Murcia, Espinardo, 30100 Murcia, Spain; 2I-Food Group, IIAD, Universitat Politècnica de València, Camino de Vera s/n, 46022 Valencia, Spain

**Keywords:** extrusion cooking, pea, lucerne, spinach, Chlorella, analogues, technological properties, sensory analysis

## Abstract

There is currently a growing trend towards the consumption of vegetable protein, even if it shows some deficiencies in essential amino acids. It has been driven by consumer passion for health and wellness, environmental sustainability, animal welfare and the flexitarian lifestyle. However, the formulation of plant protein food analogues to meat products is complicated by the technological properties of isolated plant protein. One of the processes used to improve these properties is the texturisation of the protein by extrusion, as well as the use of other plant materials that can enrich the formulation. Therefore, the aim of this study was to evaluate the effect of pea protein (PP) enriched with lucerne (L), spinach (S) and Chlorella (C) in powdered and texturised forms on the physicochemical properties and extrusion parameters, and to evaluate its technological and sensory quality as a meat analogue in vegetal hamburgers. Texturisation reduced the number of soluble components released, thus reducing the molecular degradation in extruded material. The texturised samples were significantly (*p* < 0.05) less hygroscopic than the non-textured samples. Once the properties of the powder and texturised had been analysed, they were used to prepare vegetal hamburgers. The addition of vegetable-enriched texturised samples with high chlorophyll content led to more intense colour changes in the vegetal hamburgers during cooking, with PP+C providing the darkest colouring, and also resulted in a final product more similar to a traditional meat hamburger, with higher overall and meat odour/flavour intensity, hardness, juiciness and chewiness, and less legume and spice odour and flavour. Overall, texturisation improved the technological properties of the enriched protein isolate, allowing for more efficient production of vegetal hamburgers.

## 1. Introduction

Climate change is a response to energy imbalances in the climate system that are directly or indirectly produced by human activity [1]. Rapid global warming results from the accumulation of heat-trapping greenhouse gases (GHGs) such as carbon dioxide, methane and water vapour. Animal agriculture is a significant contributor to climate change; approximately 25–30% of global GHG emissions and 70–85% of the worldwide water footprint are due to animal agriculture [2]. Due to animal metabolism, the conversion of plant protein into animal protein is inherently inefficient, so meat production is responsible for a disproportionate share of land use, freshwater depletion, global warming and biodiversity loss [3]. For example, around 7 kg of grain are required to produce 1 kg of beef, 4 kg of grain for 1 kg of pork and 2 kg of grain for 1 kg of poultry [4]. Animal agriculture does not efficiently use valuable land resources because a large portion of the acreage of farmland is devoted to producing livestock feed rather than human food directly. This has a major environmental impact, with repercussions on food production conditions. Therefore, the last three years were banner years for plant-based meat alternatives, with their dramatic growth outpacing that of their conventional counterparts [5]. US consumers perceived plant proteins to be healthier than animal proteins. One of the reasons for the positive developments is that the plant-based meat category is no longer largely viewed as limited to vegans and vegetarians. Another reason, especially for mainstream consumers, is the success of current plant-based meat products to replicate the appearance, taste, texture, aroma and mouthfeel of real meat [5]. However, the production of vegetable protein-based products is complex, as meat-like products have different technological characteristics and require texturisation to improve them. Meat analogues often have limited sensory properties, mainly in terms of texture, so the response of proteins to processing plays an important role in the structure and nutritional value of analogues [5,6]. There are various techniques for protein texturisation, but the most effective is extrusion [5]. Extruded texturised proteins have a structure that can withstand hydration and a significant amount of shear when in a fully hydrated state, similar to what would be expected from meat products [6].

The increase in consumption of plant-based meat indicated above [5,7,8,9,10] was most marked in hamburger-type products [11]; thus, the supply of hamburgers of vegetable origin has increased. The primary protein source in these products is soya [12]. Due to their allergenic power, in addition to soy proteins, proteins from the legume family have gained importance as a protein source for meat analogues such as peas, beans and chickpeas as they are rich sources of nutritious and accessible protein (around 20% dry weight) [7]. However, although vegetables are abundantly available and low-cost sources of protein, the complete elimination of animal protein means a significant deficiency in high-quality vitamins (A, B12, D), calcium and essential amino acids (lysine and methionine), which have important implications for the body. Pea protein is rich in the essential amino acids lysine and branched-chain amino acids, but it is still deficient in methionine and vitamins [13,14]. Although analogues should not only imitate the nutritional profile of meat, in addition to its appearance and texture, mouthfeel and cookability are especially important for good marketability [15], and these aspects should also be considered. To provide a good source of vegetable protein, different options have been studied, such as the use of pea starch as a cheap, non-allergenic alternative to soy and for its ability to improve the texture and especially the viscosity of the products [16], but above all, in the case of vegetal hamburgers, it is more attractive to use it in the form of a textured protein that can imitate the fibrillar structure of meat muscle [17].

A current focus of the food industry is the fortification of products to increase biodiversity based on alternative proteins, resulting in safer and nutritionally improved foods with high sensory quality. It has been shown that using other plant components, such as fibre, can help to improve the technological properties of meat emulsions [9]. Dietary fibres are mainly non-digestible polysaccharides that can enhance the appearance and texture of meat products, as they increase the water-holding capacity and the stability of emulsions during storage [18]. Furthermore, the use of additional ingredients, not only as sources of fibre but also as nutrients, to improve or increase the nutritional value of these textured products, especially the amino acid and mineral profile, is a promising avenue for the creation of novel foods [14]. Microalgae are used commercially to improve the nutritional value of foods, as they are rich in proteins, pigments, vitamins, fatty acids and polysaccharides, which have potential health benefits [7,19]. Therefore, they are a good source of protein for meat analogues, particularly *Chlorella vulgaris*, which is comparable to beef due to its similar content of the amino acid methionine [20].

Recently, to achieve good nutritional and sensory quality, pea and oat protein blends have been used to produce extruded meat analogues with complete amino acid profiles [17,21,22]. Therefore, plant protein-based meat analogues can be designed to be healthy; their enrichment with dietary fibre and plant-based amino acids can provide these vegetal hamburgers with improved sensory and nutritional profiles not found in the original meat hamburgers [11]. To meet market demands, the additional use of Chlorella and other sources of plant fibres, such as cheaper plants or forage legumes, e.g., lucerne, is presented as a promising avenue to enrich meat analogues due to their valuable nutritional composition [23]. For this reason, the aim of this study was to evaluate the effect of pea protein enriched with lucerne, spinach and Chlorella, both in powder and texturised forms, on the physicochemical properties and extrusion parameters, as well as to evaluate its technological and sensory qualities as a meat analogue in hamburgers.

## 2. Materials and Methods

### 2.1. Raw Materials

Pea protein (PP) was supplied by Roquette S.L. (Valencia, Spain), and lucerne powder (L) was obtained by freeze-drying the leaves and stems of plants grown in Aldehuela (Teruel, Spain). Spinach powder (S) was purchased from Sosa Ingredients S.L. (Cataluña, Spain), and Chlorella powder (C) was supplied from Alga Energy S.A. (Madrid, Spain).

### 2.2. Mixtures and Preparation of Texturised Protein

PP was mixed with 1% of L, S and C to obtain powder mixtures of PP+L, PP+S and PP+C, respectively. These mixtures and the PP were hydrated until their water content was 30% prior to extrusion. Then, samples were introduced into a single-screw Kompaktextruder KE 19/25 extruder (Brabender, Duisburg, Germany). The conditions used for extrusion (Figure 1) were a 3:1 compression ratio, a dosing speed of 18 rpm (feed rate range, 1.34 kg/h), a nozzle 3 mm in diameter, 150 rpm of screw rotation and 40, 80, 120 and 120 °C temperature barrel sections.

Barrel temperatures (T_1_ and T_2_), melted pressure (P), screw speed and motor torque were registered using Extruder Winext software (Brabender). Extruded products were immediately dried at 25 °C for 18 h. Dried samples were stored in polyethylene bags at room temperature (25 °C) and used for further analysis.

### 2.3. Physicochemical Properties of Powder and Texturised Protein

#### 2.3.1. Water Content (x_w_)

The water content of the samples was determined by drying the samples until they reached a constant weight at 105 °C in a vacuum oven [24]. From the water content of the samples after extrusion and drying, water losses for the processes of extrusion and drying were calculated.

#### 2.3.2. Water Solubility Index (WSI) and Water Absorption Index (WAI)

The water solubility index and water absorption index were determined using the method of Singh and Smith [25] and calculated according to Igual et al. [23]. The sample was dispersed in distilled water. After stirring, the sample was centrifuged at 3000× *g* for 10 min. The supernatant was decanted to determine its dissolved solid content, and the sediment was weighed. The swelling index (SWE) was measured following the protocol of Roberson et al. [26] based on the bed volume technique. Samples were weighed and transferred to a graduated test tube, and then distilled water was added. Test tubes were maintained for 18 h at ambient temperature (25 °C). The bed volume was measured and expressed as mm of swollen sample per g of the dry initial sample. The fat adsorption index (FAI) was determined according to Navarro-González et al. [27], with minor modifications [28]. Samples were stirred for 30 s every 5 min for 30 min with sunflower oil. Then, they were centrifuged at 1600× *g* for 25 min. Free oil was decanted, and FAI was expressed as *g*_oil_/*g*_sample_.

#### 2.3.3. Hygroscopicity (Hy)

Hygroscopicity was determined according to Cai and Corke [29]. The samples remained in contact with an atmosphere with a relative humidity of 81% at 25 °C in a container with saturated Na_2_SO_4_ solution. Samples were weighed after 1, 4 and 7 days, and hygroscopicity was expressed as g of water gained per 100 g of dry solids.

#### 2.3.4. Instrumental Colour

The colour of the powder samples was measured with a standard D65 illuminate and 10° visual angle (Konica Minolta CM-700d colourimeter, Tokyo, Japan). A reflectance glass (CR-A51, Minolta Camera, Japan) was placed between the sample and the colourimeter lens. The measurement window was 6 mm in diameter. The results were expressed using the CIELab system. Chroma, C* (saturation) and hue angle, h*. The total colour difference between the powdered and texturised samples (ΔE_1_) and between the PP and enriched samples (PP+L, PP+S and PP+C) (ΔE_2_) were calculated for both powder and texturised protein.

All analyses were performed on the powdered and texturised samples.

### 2.4. Formulation and Preparation of Hamburgers

Vegetal hamburgers were made as a model product system to evaluate enriched powders and texturised protein as a source of protein in the production of meat analogues.

The hamburgers were produced according to the method described by Botella-Martínez et al. [11] and Bakhsh et al. [30] with slight modifications. These were produced in the pilot plant of the Joint Research Centre (CIAVYS-VITALYS, University of Murcia), using 38% (*w*/*w*) hydrated pea protein (30% water), 50% (*w*/*w*) methylcellulose solution at 3% (E-461, Texturalia, Asturias, Spain), 2% (*w*/*w*) dry spices (Hacendado, Valencia, Spain) and 10% (*w*/*w*) vegetable fat (extra virgin olive, Hacendado, Valencia, Spain). Methyl cellulose was established as a stabiliser to form a more resistant and thermally stable gel [31]. Four vegetal hamburger formulations were developed, varying the source of enrichment (lucerne, spinach or Chlorella) for both protein formats (Table 1). The protein concentration was used to determine the percentage of other ingredients added to the formulation.

For the preparation, 3% methylcellulose was first conditioned, hydrated at 4 °C for 24 h for better dispersion and dissolved at 50–55 °C. Once conditioned, the rest of the ingredients needed for the preparation were weighed. The protein source was added to the methylcellulose, and the first homogenisation was carried out. The spices and olive oil were then added to the mixture and mixed by hand until a homogeneous mass was obtained. The blends of the four formulations were then manually moulded (33.3 g final vegetal hamburger blend); therefore, three vegetal hamburgers were prepared from each formulation (100 g). All formulations were produced in quadruplicate per production day. Once moulded, they were left to rest at 4 °C for 24 °C. Finally, they were vacuum-packed and frozen at −18 °C in a freezing chamber until analysis. The samples were named according to protein source P (powder) and T (texturised protein): PP (pea protein), PP+L (pea protein with 1% lucerne), PP+S (pea protein with 1% spinach) and PP+C (pea protein with 1% Chlorella). Three batches of each formulation were made on different days in separate weeks following the same process (replicates).

### 2.5. Physicochemical Properties of Formulated Hamburgers

Physicochemical analyses were carried out on fresh (colour- and water-holding capacity) and cooked (cooking loss, colour and instrumental texture) samples. The burgers were previously thawed at 4 °C for 24 h.

#### 2.5.1. Instrumental Colour

The colour of the raw and cooked vegetal hamburgers was measured as described in the previous Section on the sample surface from three randomly chosen spots. The following CIELAB colour coordinates were obtained: lightness (L*), redness (a*) and yellowness (b*). Psychophysical quantities, namely, °hue (h*) and chroma (C*), were calculated from the colour coordinate by using Equations (1) and (2) [32]:(1)$$C^{*}={\left(a^{*2}+b^{*2}\right)}^{1/2}$$
(2)$$h^{*}=\tan^{-1}\left(b^{*}/a^{*}\right)\;$$

#### 2.5.2. Water-Holding Capacity (WHC)

Measurement of water-holding capacity was carried out according to the technique of Grau and Hamm [33]. This technique measures the water released by a sample after subjecting it to a pressure of 1 kg. To measure the % WHC, a 0.3 g sample was placed on a Whatman No. 540 paper filter and subjected to pressure for 10 min between two Petri dishes. The Whatman paper was kept in an environmental microchamber at 84% relative humidity. The weights of the Whatman papers and samples were noted. After the pressure was applied, the weight of the Whatman paper was measured to determine the percentage of released water by the sample. It was calculated using Formula (3):(3)$$\mathrm{WHC}\;\left(\%\right)=100-\left[\left(\frac{\mathrm{Pfinal}-\mathrm{Pinitial}}{\mathrm{Psample}}\right)\times 100\right]$$
where:

Pfinal = final weight of the paper (g)

Pinitial = initial weight of the paper (g)

Psample = sample weight (g)

#### 2.5.3. Cooking Loss (CL)

The cooking loss of the enriched pea protein-based burgers was calculated based on weight differences according to the method described by Wi et al. [8]. Cooking conditions were set at a temperature of 180 °C on a griddle (Velox CG-1S, Silesia, Spain, Barcelona) for 10 min until the meat analogue reached an internal temperature of 80 °C. The internal temperature of the meat analogue was measured with a penetration probe. After cooking, the samples were cooled to room temperature for 30 min. The CL was calculated as the percentage weight difference between the mass before and after cooking, according to Equation (4):(4)CL (%)=(W1−W2)/W1×100
where:

W1: weight of meat analogue dough (g).

W2: weight of cooked meat analogue (g).

#### 2.5.4. Texture Profile Analysis (TPA)

Texture profile analysis of the vegetal hamburgers was carried out using a Texturometer CT310K (Brookfield CNS Engineering Labs. Inc., UK, Harlow) with TexturePro CT V1.8 software, according to the procedures described by Lee and Hong [34] and Wi et al. [8]. Samples were cut into 2 cm diameter × 2 cm wide cylinders and conditioned at 23 °C. A double compression cycle test was performed with a compression of up to 50% of the height of the original portion with a cylindrical probe 10 mm in diameter (TA 10) and a 25 kg load cell. The force–time deformation curves were obtained with a velocity of 2.5 mm/s and a trigger point of 5 g. The parameters which we determined were hardness (g), adhesiveness (mJ), chewiness (mJ), gumminess (g), cohesiveness, elasticity (mm) and resilience (J/m^3^).

All experiments and analyses were carried out in triplicate, with the exception of the texture profile, which was performed in quadruplicate.

### 2.6. Sensory Analysis of Formulated Hamburgers

Six panellists, four women and two men, belonging to the panel of sensory experts from the Food Technology Department of the University of Murcia (Spain) were selected based on their previous experience in sensory analysis. A total of 4 1.5 h theoretical–practical training sessions were carried out with the panellists to familiarise them with the product and to identify the descriptors that best matched the products and the ranges of each descriptor according to ISO 8586:2012 [35]. The list of sensory attributes, definitions and scale references is reported in Table 2. To assess the sensory attributes of the fortified vegetal hamburger preparations, a QDA (quantitative descriptive analysis) test was performed using a 10-point unstructured scale [36]. For the descriptive sensory analysis, samples were cooked on a griddle (Velox CG-1S, Silesia, Barcelona, Spain) at 180 °C for 10 min until an internal temperature of 80 °C was reached (T200 portable thermometer, Digitron Instrumentation Ltd., Hertford, UK). The cooked samples were cut into 2 × 2 cm pieces, wrapped in aluminium foil, coded with a three-digit numerical code, and kept in a sand bath at 60 °C until tasting [37]. The order of sample presentation within each session was balanced to account for order and carryover effects [38]. Each panellist evaluated 3 samples from each preparation (4 elaborations × 2 pea protein formats (powder and texturised) × 3 replicates) in a standardised room according to [39].

### 2.7. Statistical Analysis

Analysis of variance (ANOVA), using Statgraphics Centurion XVII Software, version 17.2.04, with a confidence level of 95% (*p* < 0.05), was applied to evaluate the differences among samples. The method used to discriminate between means was Fisher’s least significant difference procedure. Data are reported as means and standard deviations for each variable. A correlation analysis was conducted between the properties of proteins and the properties of vegetal hamburgers formulated from these proteins, with a 95% significance level d (Statgraphics Centurion XVII).

All sensory evaluation data on the hamburgers were analysed with the statistical software package SPSS 28 (SPSS, Chicago, IL, USA). The attributes were analysed individually by univariate complete factorial analysis of variance using the general linear model (GLM) procedure. Considering the effects of different pea protein formats (powder and texturised) and formulations (PP, PP+L, PP+S and PP+C) as a fixed source of variation, sensory attributes were considered the dependent variable, and replicates, panellists and sessions were adjusted as random effects. The effects of format (FP), formulations (FM) and the interaction of both factors (FP × FM) were evaluated at *p* < 0.05. The means, standard error of the mean (SEM) and between-sample differences (*p*-value) of the scores for each strategy were predicted from each model. Between-mean comparisons of parameters were performed using Tukey’s test, and significance was detected at the 0.05 level.

## 3. Results

### 3.1. Protein Texturisation

Table 3 shows the process control parameters for texturing protein mixtures and PP. Barrel temperatures (T_1_ and T_2_) and melt pressure (P) were monitored during extrusion. Specific mechanical energy (SME) can be defined as the energy required to produce 1 g of extrudate [40]. It was calculated [41] from torque (C, N m), screw speed (V, rad s^−1^) and the mass flow rate (Q, g s^−1^) with Equation (5).
(5)$$\mathrm{SME}=\frac{C\times V}{Q}$$

The addition of L, S and C caused a significant (*p* < 0.05) increase in SME. Other studies have shown that feed composition significantly affects SME. Specifically, they indicate that a higher carbohydrate content in mixtures caused higher SME [40]. In the formulation, samples containing L, S or C have more carbohydrates than PP (only protein). SME values order the enriched samples from highest to lowest: S > C > L. This is likely due to the composition of the raw material, since, as indicated above, the feed composition exhibits a strong effect on SME [42].

T_1_, T_2_ and P showed no significant (*p* > 0.05) differences when L, S or C were added into the mixtures.

The appearance of the samples obtained after the texturisation process can be seen in Figure 2. They all show fibrous structures typical of texturised protein [5]. The wetted proteins were plasticised in the extruder barrel during extrusion by applying heat, pressure and mechanical shear. The plasticised dough was then pushed through the die openings, during which the water in the dough partially evaporated, and protein molecules rapidly aligned to generate fibrous textures [43]. Native protein structures were altered in response to extrusion energy, resulting in denaturation, conformational changes and alterations in technological properties [5,10].

The wet protein was extruded and dried to obtain the samples, as described in Section 2.3. During extrusion, water was evaporated from the sample, and during the process of drying, the sample suffered water losses. Figure 3 shows the water losses in both stages (extrusion and drying) before the final product was obtained. No significant (*p* > 0.05) differences in water losses during extrusion were observed. The total water losses of PP+L showed significant (*p* < 0.05) differences due to drying, probably because the water was less bound to the matrix than in the other samples.

The moisture and hygroscopic character of a sample are indicators of the stability of the samples. Table 4 shows the mean values and deviations of the moisture and hygroscopicity of the samples. In the protein texturisation process, water is added before the texturisation process, which is then lost in the stages mentioned above. However, the texturised samples had significantly (*p* < 0.05) higher water contents than untextured powder samples. Comparing the powdered samples, PP+C was the least moist. In the case of the texturised samples, the least moist was PP+L, as it lost more water than the others in the texturisation process (Figure 3). When the samples were stored for 1d in an environment with 81% relative humidity, the textured samples took up approximately half of the water content from the environment that the untextured powder samples did. They are, therefore, more stable, as less water uptake makes them less favourable to degradation reactions. The hygroscopicity values of the textured samples were significantly (*p* < 0.05) lower than the non-textured samples for the three studied times.

Figure 4 shows the behaviour of the samples in contact with water (WAI, WSI and SWE) and oil (FAI). The WAI values were similar for textured and non-textured samples. PP+L showed significant differences as a function of texturisation, with a higher WAI in the texturised sample. PP+C showed the opposite trend, with the texturised sample showing the lowest WAI. In terms of formulation, the non-texturised powder sample with the highest WAI was PP+C; however, it was PP+L for the texturised samples. WSI refers to water-solubilised components released during texturisation [44]. Texturisation reduced the number of soluble components released, as indicated by WSI values in Figure 4, thus reducing the molecular degradation in samples because there are fewer soluble particles to react with water and degrade the matrix. Protein solubility is often a vital indicator of the degree of protein texturisation [45]. After extrusion cooking, the protein is thermally denatured with a series of cleavages and aggregations, decreasing the soluble protein content. Thus, texturised proteins with lower solubility than their native counterparts are often observed [10,44]. This effect was observed in maize samples fortified with cowpea as a protein source [46]. Figure 4 also shows the SWE values, which decreased significantly (*p* < 0.05) when the samples were textured. This decrease in SWE was observed in all samples equally.

Texturisation of the samples led to a significant (*p* < 0.05) decrease in FAI in all samples (Figure 4), which is in agreement with the results obtained by Sotelo-Díaz et al. [46] that FAI was reduced with the extrusion of corn samples with cowpea. Extrusion processing caused a decrease in the number of available hydrophobic sites in the samples. This may be due to the higher extrusion temperature. When protein denaturation takes place together with aggregation, interactions among hydrophobic groups occur during the formation of aggregates, leading to an overall decrease in the hydrophobicity of the product. Owing to this interaction, blocking of the possible hydrophobic sites that can react with oil may occur [47]. On the other hand, adding L, S and C to untextured PP powder increased FAI, as its composition includes fibres and carbohydrates that facilitate oil absorption.

Table 5 shows the colour coordinates and the differences between samples due to the formulation effect or texturisation effect. Texturisation decreased significantly (*p* < 0.05) Texturisation decreased significantly (*p* < 0.05) L* coordinate and increased significantly (*p* < 0.05) b* and C*. The samples darkened and lost their yellowish tones after texturisation. In terms of the colour differences induced by texturisation (ΔE_1_), the most significant differences were observed in PP, followed by PP+S and PP+C, and, finally, PP+L. All of these differences were perceptible to the human eye, as they consisted of more than 3 units of change [48]. The colour perception of the samples can be seen in Figure 5, and the effect of texturisation on the colour appearance of the samples is evident. The addition of L, S or C to the protein powder decreased L* and a*. Concretely, the addition of L or S to the protein powder showed no changes perceptible to the human eye in the samples (ΔE_2_ > 3 units, [48]). However, using C increased the perceptibility of the colour differences (ΔE_2_ > 3 units). The effect of enrichment with L, S or C in the textured samples caused perceptible changes compared to PP (ΔE_2_ > 3 units).

### 3.2. Application as a Substitute for Meat in Hamburgers

Table 6 shows the results of the instrumental colour- and water-holding capacities (WHC) of the different hamburger formulations. The CIELab colour showed significant differences (*p* < 0.05) between formulations (PP, PP+L, PP+S and PP+C) in both formats (powder and texturised), with the PP+C sample having the lowest values for all colour coordinates except for the hue angle, which was higher (*p* < 0.05). This formulation produced darker final products, with lower lightness (L*) and C* and higher h* values, with less reddish and yellowish and more greenish colourations. This decrease in the L* value and these colourations was mainly due to the different colours of the chlorophyll pigments (green and blue-green) of the microalgae species used to enrich the pea protein extract compared to the colour of the legume [19]. The formulation with pea only (PP) had the highest score for brightness and reddish colour (a*), and the lowest for hue (h*), followed by the PP+S formulation. For the b* and C* coordinates, both formulations showed similar values (*p* ≥ 0.05). In general, the colour changes in the formulations were closely related to the chlorophyll content of the ingredients used in the fortification (lucerne, spinach and Chlorella) of each formulation. The main source of chlorophyll was algae, followed by cereals such as lucerne, and, to a lesser extent, green leafy vegetables such as spinach [49]. Similar results were obtained by Žugčić et al. [7], who observed how adding Chlorella or spirulina into the extrudate mixtures affected the colour parameters. The higher the concentration of chlorophyll pigments, the lesser the brightness and reddish colouring of meat hamburgers.

Regarding the extrusion process, significant changes (*p* < 0.05) were observed in all colour coordinates, except h* for the enriched formulations and L* for the PP+ L and PP+C formulations (*p* ≥ 0.05). There were significantly more differences in L*, a*, b* and C* in the vegetal hamburgers produced with powder than with texturised protein, and fewer differences in h* in the PP formulation. Therefore, extrusion affected the heat-sensitive pigments of the samples. Colour is highly dependent on raw materials and extrusion process parameters, as the high temperature used during the extrusion process involves pigment degradation [50]. Consequently, at a high extrusion temperature of 170 °C, Igual et al. [51] found a decrease in the L*, a*, b* and C* values and an increase in the tone of corn extrudates added with lucerne, as was the case in our study. In addition, the colour of extruded products is also affected by food moisture. In the work of Selani et al. [52], the researchers observed a decrease in lightness and redness with higher moisture content in extrudates with pomace.

Water-holding capacity (WHC) is a fundamental property of meat products. It has a significant influence on the performance and sensory acceptability of the product, as it measures the ability of the protein to retain water by capillary and tensile force and to form the protein gel network [8], so it is of great importance in meat analogues. Among the formulations (PP, PP+L, PP+S and PP+C), significant differences (*p* < 0.05) were observed for both formats. PP+S and PP+C showed higher water-holding capacities in the powder samples, as PP+L and PP+S did in the texturised ones; these results coincide with those obtained for the water absorption index (WAI). The WHC depends on the amino acid composition and protein–polysaccharide interactions, which are based on electrostatic forces, hydrogen bonds and the microstructure of the binding agent, but are also related to the presence of fibres [53]. The addition of fibre has been shown to improve the water-holding capacity and stability of emulsions during storage [9,18]. Thus, our results indicate this, since the ingredients used for the fortification of the formulations were rich sources of fibre, especially lucerne and spinach, with fibre contents of 3.5 (forage) and 27.3 (leaf meal) g/100 g for lucerne [54] and 6.3 g/100 g for spinach [55].

Although differences between formulations were observed, these were not very pronounced, with all formulations obtaining high WHC values around 91.99–93.76% in the powder and 82.23–86.30% in the texturised samples. Similar results were observed by Bakhsh et al. [30], who studied the technological properties of different concentrations of methylcellulose in soybean patties. They found that the use of high concentrations of methylcellulose (3–4%) reduced water release and cooking losses, which increased the water-holding capacity of the samples. Methylcellulose has high binding and moisture retention capacities, as it is a potent gelling and thickening agent that increases volume and improves the texture of processed meat and meat analogues [56].

Regarding the source of protein application in vegetal hamburgers (powder and texturised), the results showed that burgers made with powder had a higher water retention capacity in all formulations (*p* < 0.05). Protein isolate powder can absorb more water in its native globular structure than during protein denaturation after extrusion [57]; similar results were observed by Kaleda et al. [22] in oat and pea protein meat analogues.

**Table 6 foods-12-01303-t006:** Mean values (and standard deviations) of colour coordinates and WHC of fresh vegetal hamburgers manufactured using enriched pea protein (PP, PP+L, PP+S and PP+C). PP, pea protein; PP+L, pea protein with lucerne; PP+S, pea protein with spinach; PP+C, pea protein with chlorella.

	Sample	L*	a*	b*	C*	h*	WHC
Powder	pp	58.14 (1.15) ^aA^	8.22 (0.26) ^aA^	13.19 (0.83) ^aA^	15.54 (0.80) ^aA^	58.06 (1.27) ^dB^	92.33 (1.12) ^bA^
	PP+L	55.10 (1.54) ^bA^	4.55 (0.07) ^cA^	12.29 (1.37) ^aA^	13.11 (1.29) ^bA^	69.57 (1.91) ^bA^	91.99 (0.61) ^bA^
	PP+S	57.41 (0.83) ^aA^	6.43 (0.19) ^bA^	13.27 (1.05) ^aA^	14.75 (0.99) ^aA^	64.10 (1.47) ^cA^	93.76 (0.99) ^aA^
	PP+C	51.78 (0.99) ^cA^	2.13 (0.16) ^dA^	9.13 (0.90) ^bA^	9.38 (0.88) ^cA^	76.81 (1.51) ^aA^	92.91 (0.67) ^abA^
Texturised	pp	56.67 (0.57) ^aB^	6.85 (0.48) ^aB^	11.95 (0.75) ^aB^	13.78 (0.78) ^aB^	60.17 (1.81) ^dA^	83.39 (0.43) ^bB^
	PP+L	55.29 (1.42) ^bA^	4.33 (0.23) ^cB^	11.39 (0.64) ^abB^	12.19 (0.56) ^bB^	69.17 (1.82) ^cA^	86.30 (1.82) ^aB^
	PP+S	54.69 (0.44) ^bB^	5.04 (0.27) ^bB^	10.28 (1.27) ^bB^	11.45 (1.23) ^bB^	63.75 (2.02) ^bA^	85.04 (0.43) ^abB^
	PP+C	51.97 (1.09) ^cA^	1.85 (0.12) ^dB^	7.89 (1.20) ^cB^	8.10 (1.19) ^aB^	76.62 (1.68) ^aA^	83.23 (1.91) ^bB^

L*: lightening, a*: red-green; b*: yellow-blue, C*: chroma, h*: °Hue, WHC: water-holding capacity. For each parameter, the same small letters indicate homogeneous groups established by ANOVA (*p* < 0.05) comparing formulations (PP, PP+L, PP+S and PP+C) in powder and texturised protein samples. For each sample (PP, PP+L, PP+S and PP+C) and parameter, the same capital letter indicates homogeneous groups established by ANOVA (*p* < 0.05) comparing powder and texturised samples.

The physical and mechanical properties of the cooked patties are presented in Table 7. Regarding the CIELab colour of the cooked hamburger formulations, all colour coordinates showed similar trends to those described for the fresh vegetal hamburgers (*p* < 0.05), with the formulation enriched with Chlorella showing the lowest values for L*, a*, b* and C* and the highest values for h* (*p* < 0.05). During the extrusion process, changes were also observed in the colour coordinates, with the burgers made with powder showing the highest scores for these coordinates (*p* < 0.05). Chroma being the only colour parameter that showed significant differences between the two protein formats in all the hamburger formulations (PP, PP+L, PP+S and PP+C).

Figure 6 shows digital photographs of the same vegetal hamburger formulations before and after cooking. Although the cooked and fresh samples showed similar trends, the cooking process was evident in the colour coordinates, with the cooked vegetal hamburgers showing lower lightness (L*), yellow (b*) and hue (h*) in their colouring and higher reddish (a*) colouring, while the chroma scores were very similar to those of the fresh vegetal hamburgers. During cooking, the high temperatures can degrade the pigments in the samples, especially chlorophyll, which is very sensitive to high temperatures. Heat disintegrates the cellular structure of the chlorophyll and leaves the pigment exposed to various enzymatic and non-enzymatic reactions that brown the product by conversion to pheophytins (devoid of their magnesium ion in the porphyrin ring) [58]. Hence, cooked patties have lower brightness values and less green colour than fresh vegetal hamburgers. In addition, other reactions occur during cooking, such as protein denaturation and aggregation, water evaporation and Maillard reactions, which could contribute to the changes in the samples [59]. Maillard reactions occur during heating between amino acids and reducing sugars, causing caramelisation of the samples and giving an orange and brownish colouring [60].

In meat products, the reactions of water evaporation, denaturation and protein aggregation also affect the ability of the emulsion to bind water and fat, and are, therefore, responsible for cooking losses (CL). Regarding CL, significant differences were found between the formulations for both protein formats (*p* < 0.05), with the PP+S and PP+C formulations showing fewer losses in powder and PP+L and PP+S in texturised samples. These results coincide with those obtained for WHC, as it is associated with the weight losses of meat products during cooking [8]. Therefore, the formulations that presented greater WHCs were the ones that showed the lowest cooking losses. Angiolillo et al. [13] established that adding fibres to non-meat protein ingredients reduces weight loss, as some dietary fibres can strengthen the connections between the different components of the matrix by preventing water diffusion after cooking [61].

Among the protein formats, it was observed that powders showed the lowest cooking losses compared to texturised proteins (*p* < 0.05) at around 10–14% and 25–28%, respectively. These differences may have to do with the greater ability of protein isolate powders to absorb water into their structures than proteins denatured during extrusion [57]. In general, with texturised vegetable proteins, the cooking losses obtained were higher than those reported by other authors for commercial and non-commercial soy or pea vegetable burgers, which were around 10–15% CL [10,30,59]. These lower losses could be attributed to high concentrations or a mixture of binding agents during the production of vegetal hamburgers, such as gums, wheat gluten and corn starch, and not only methyl cellulose as used in this study; these contribute to emulsion stability [59].

In terms of the textural properties (TPA) of the cooked vegetal hamburgers, there were no significant differences (*p* > 0.05) for the parameters of hardness, adhesiveness, resilience, cohesiveness, elasticity or gumminess between any of the formulations made with protein powder. For those made with texturised protein, no significant differences (*p* > 0.05) were observed for the parameters of hardness, adhesiveness, elasticity or gumminess, but significant differences were observed for resilience, cohesiveness and chewiness (*p* < 0.05), with the PP formulation showing the highest values for resilience and PP and PP+S for cohesiveness. The chewiness was the only parameter affected by formulation (*p* < 0.05) in both protein formats, being higher in PP+S and PP+L formulations in the powder and in PP, PP+S and PP+C in the texturised samples. Resilience measures how fast and strong the recovery is, and cohesiveness indicates the strength of the internal bonds to hold together; in meat analogues, it is the proteins that mainly contribute to the three-dimensional internal structure through hydrophobic interactions, and the structure is stabilised by hydrogen and disulphide bonds [62]. The moisture and fibre content of vegetal hamburgers are factors responsible for their mechanical properties; decreasing the moisture content and increasing the fibre content of the food would increase its toughness and chewiness [8]. López-López et al. [63] showed that the addition of seaweed extracts (*H. elongata*) rich in fibre leads to an increase in the hardness and chewiness values, while it reduces the elasticity and cohesion of sausages. In the work of Zhou et al. [59], they obtained similar results to ours in terms of hardness, cohesiveness, gumminess and adhesiveness in commercial vegetable burgers, with a hardness of 440 g for vegetal hamburgers made with textured pea protein and 1300 g for those made with soybean flour. However, resilience, elasticity and chewiness had higher scores than in our study [59]; elasticity has been found to correlate with higher protein concentrations and, thus, greater chewiness [17].

Among protein formats, the powder had the highest values for hardness, resilience, cohesiveness, elasticity, gumminess and chewiness (*p* < 0.05). The toughness of the vegetal hamburgers appears to be related to their cooking losses and water-holding capacities, as toughness tends to decrease as the moisture content of the samples increases [8]. In particular, the cooking losses of the protein powder samples were lower than those of the texturised samples. This may be due to their denser structure; as they had higher oil absorption capacities, they could better capture the oil incorporated in the emulsion and form a network more resistant to deformation during cooking. Protein–lipid and carbohydrate–lipid interactions stabilised the structure more effectively [64].

The results of the sensory analysis are presented in Table 8. Statistically significant differences (*p* < 0.05) were observed in all the sensory attributes analysed, both for the effects of the protein format (PF: powder and texturised) and hamburger formulations (FM: PP, PP+L, PP+S and PP+C) and for the interaction between them (PF × FM). However, significant differences were not found for the fibrousness attributes of the two formulations, nor for the interactions of FP and FM (*p* ≥ 0.05) or for the hardness of FP x FM (*p* ≥ 0.05). In terms of the appearance of the burgers, it could be seen that as the chlorophyll content increased, the colouring of the formulations darkened. PP+C showed the darkest colour, followed by PP+L and PP+S, with the formulation without enrichment (PP) showing the colouring most characteristic of a pea-based hamburger. As seen in the instrumental colour coordinates, this is mainly due to the higher concentration of chlorophyll pigments in Chlorella [7]. Regarding brightness, the PP+C sample had the highest values for this attribute (*p* < 0.05) compared to the other formulations, which did not differ from each other (*p* ≥ 0.05). Brightness is determined by the amount of water or fat on the surface of the food that is capable of reflecting light [65]; the formulation enriched with Chlorella had the lowest water retention capacity and the highest cooking losses, so when cooked, it released more water to the surface of the hamburger, providing a gloss effect. No significant differences were observed in the fibrousness between formulations (*p* ≥ 0.05), with the panellists giving the same fibrousness score to all formulations. Regarding the effect of protein format, the texturised samples showed higher colour tones due to Maillard reactions during extrusion [58], as well as higher brightness and fibrousness values, than burgers formulated with protein powder (*p* < 0.05), which did not have fibrous structures, as can be seen in Figure 7. Extrusion cooking of vegetable proteins allows us to obtain a fibrous matrix that mimics the fibrillar structure of meat muscle [17].

For the odour attributes, it was observed that the formulation with microalgae (PP+C) had general, vegetable, and meat odours of higher intensity and legume and spiced aromas of lower intensity (*p* < 0.05). In contrast, the PP+L formulation had the lowest scores for overall and meat odour, followed by the PP formulation, which also had the lowest meat odour and vegetable odour scores. The highest scores for legume and spiced aroma were shown by PP and PP+S (*p* < 0.05). In general, an inverse relationship between the increase in legume and spiced aromas and the decrease in odour intensity can be observed. The addition of 1% Chlorella could have contributed to softening the intensity of the volatile compounds responsible for the legume odour [17,21], which is usually not very appreciated by consumers [66] because of its stronger vegetable odour. Spices and herbs with intense, fresh scents are typically used to mask these products’ unpleasant connotations [67]. One of the most outstanding properties of Chlorella is its fresh green fruity odour and flavour, which exert an odour-neutralising effect [68]. Regarding the effect of protein format, it could be seen that the vegetal hamburgers made with texturised protein presented higher overall, spiced, vegetable and meat odour intensities, but lower legume odour intensity (*p* < 0.05), compared to protein powders. This could be related to the occurrence of Maillard reactions of amino acids/peptides during the extrusion process resulting in volatile aromatic substances, which could decrease the perception of legume odour and be responsible for the meat odour and flavour [22,69].

In terms of flavour, the PP+C formulation was perceived as the least salty and with the least umami taste (*p* < 0.05). The main umami agent is monosodium glutamate, which, like salt, is a flavour enhancer [70]. Therefore, the less salty samples were those with less of an umami taste. In contrast, the PP formulation was tastier, sweeter and, thus, the least bitter (*p* < 0.05), as sweet and umami taste receptors share a subunit, namely, the T1R receptor family (T1R1/T1R3 responding to umami and T1R2/T1R3 responding to sweet), hence the similar results for these attributes [71]. In this study, brewer’s yeast was added to enhance the umami taste [72]. Umami taste has been shown to modulate sweetness and suppress bitterness [70]. Vegetable proteins are often associated with bitterness, which may be caused by the presence of anti-nutritional factors such as saponins or phenolic compounds (tannins and catechins) [73]. De Angelis et al. [17] obtained similar results for both odour and flavour in texturised pea protein isolate analogues. As was the case for odour, the PP+C formulation had higher overall vegetable and meat odour intensity and lower legume and spiced flavour (*p* < 0.05), and it was observed that overall odour scores were higher than flavour scores. The flavour of the vegetal hamburgers made with protein powder followed the same trend as the odour, being sweeter, more bitter, more leguminous and, therefore, less salty, with less overall, umami, spiced, vegetable and meaty flavour intensity (*p* < 0.05) compared to the texturised protein. As discussed above, a more intense Maillard reaction during the extrusion process could have led to a better sensory profile of the texturised protein. To achieve the meat flavour during the thermal degradation of amino acids/peptides, starting raw materials such as free amino acids, yeast extracts and hydrolysed vegetable protein, in our case, are important [69].

For the texture attributes, the enriched formulations had the highest scores for toughness, juiciness, and chewiness and the lowest scores for adhesiveness (*p* < 0.05). As determined for instrumental texture, increasing fibre content increased toughness and chewiness [8]. As for juiciness, it is related to the moisture content of the patties (WHC and CL); formulations with a higher capacity to retain water by capillarity were juicier [60]. For the gumminess attribute, the panellists gave the highest scores to the PP and PP+S formulations (*p* < 0.05), with almost negligible values ranging between 0.17 and 0.16, respectively. For graininess, the least granular formulation was PP+C (*p* < 0.05), probably due to the lipid content of Chlorella [74], which likely helped to improve the cohesiveness of the vegetal hamburgers [7].

Regarding the format, in contrast to the values obtained for the instrumental texture, the panellists gave the textured protein the highest values for hardness, juiciness, chewiness and gumminess, and the lowest values for graininess and adhesiveness, the latter not being adhesive at all (*p* < 0.05). This is probably due to the fibrous texture obtained after the extrusion process, which was more similar to the texture of meat and less grainy than powder [17]. In general, producing vegetal hamburgers with texturised protein yielded an excellent sensory profile.

Pearson correlations were carried out to relate the properties of the textured and non-textured proteins with the properties of the vegetal hamburgers obtained from them (Table 9). WAI was not significantly (*p* > 0.05) correlated with any property measured in the vegetal hamburgers. AD also showed no significant (*p* > 0.05) correlations with the studied properties of the L-, S- and C-enriched proteins. All other parameters were significantly correlated with high coefficients. FAI, SWE and WSI showed a negative correlation with CL. CL was higher when the enriched protein showed lower values of FAI, SWE and WSI, as it has been shown that the lower capacity for swelling and absorption of fat and water by the enriched protein structure leads to lower stability of the vegetal hamburgers in the emulsion to bind water and fat, and, therefore, to higher losses during cooking [7,57]. The x_w_ of proteins was negatively related to the mechanical properties of the hamburgers and their WHC values, probably due to tensile and capillary forces of the protein gel network, as increasing moisture content hampers the protein’s ability to retain water, decreasing, in particular, its hardness, cohesiveness and chewiness [8,59]. However, x_w_ was positively correlated with CL; moister proteins resulted in vegetal hamburgers that lost more water during cooking. In general, SWE showed the highest correlation coefficients with mechanical properties. A higher protein swelling capacity produces firmer and more cohesive vegetal hamburgers, although more gummy at the same time, as these proteins have a greater capacity for swelling and oil absorption. Thus, they were able to better capture the oil incorporated during the preparation of the vegetal hamburgers and form a network that was denser and more resistant to deformation during cooking due to protein–lipid and carbohydrate–lipid interactions [64].

## 4. Conclusions

In this work, according to tasters, vegetal hamburgers with good technological and sensory properties were obtained from pea protein enriched with lucerne (L), spinach (S) and Chlorella (C). Texturisation modified the protein isolates to have more technological properties. As an ingredient, the texturised protein was less hygroscopic and less prone to degradation. The vegetal hamburgers obtained with texturised protein had high water retention capacities and, therefore, lower cooking losses, as a relationship between a greater capacity for swelling, absorption of fat by the protein structure and lower losses during cooking has been observed. However, it should be noted that the WHC was higher in the vegetal hamburgers made with powder, due to their globular native structure, than in the proteins denatured during extrusion. As for the enrichment of the texturised products with vegetables with high chlorophyll and fibre content (C, L and S), these caused more intense colour changes during cooking, with PP+C exhibiting the darkest colouring. Furthermore, adding fibre improved the WHC, texture and sensory profiles of the vegetal hamburgers, with PP+C obtaining higher overall and meat odour/flavour intensity, hardness, juiciness and chewiness, and lower legume and spiced odour and flavour, followed by PP+S and PP+L. In short, the enrichment of texturised pea protein made it possible to obtain a final product similar to a traditional meat patty due to the fibrous texture obtained after the extrusion process, which is more similar to the texture of meat and not as grainy as the samples produced with powder.

## Figures and Tables

**Figure 1 foods-12-01303-f001:**
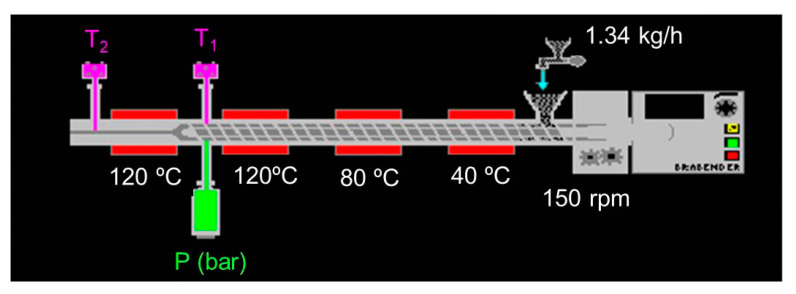
Scheme of conditions used in the extruder.

**Figure 2 foods-12-01303-f002:**
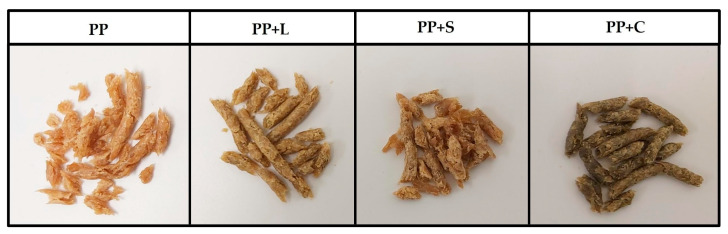
Appearance of texturised samples. PP, pea protein; PP+L, pea protein with lucerne; PP+S, pea protein with spinach; PP+C, pea protein with chlorella.

**Figure 3 foods-12-01303-f003:**
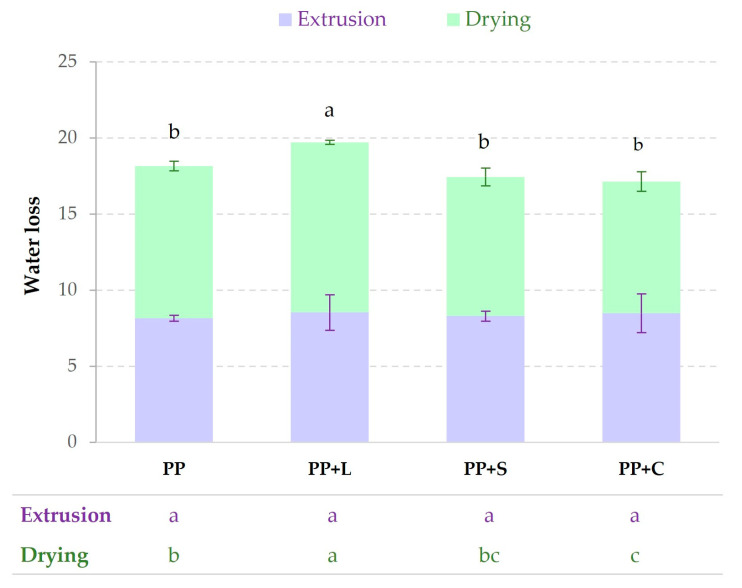
Mean values (and standard deviations) of water loss due to extrusion and drying for each sample. Table contains letters representing the homogeneous groups established by ANOVA (*p* < 0.05) for water losses at each production stage. Letters above the bars indicate homogeneous groups according to total water loss. PP, pea protein; PP+L, pea protein with lucerne; PP+S, pea protein with spinach; PP+C, pea protein with chlorella.

**Figure 4 foods-12-01303-f004:**
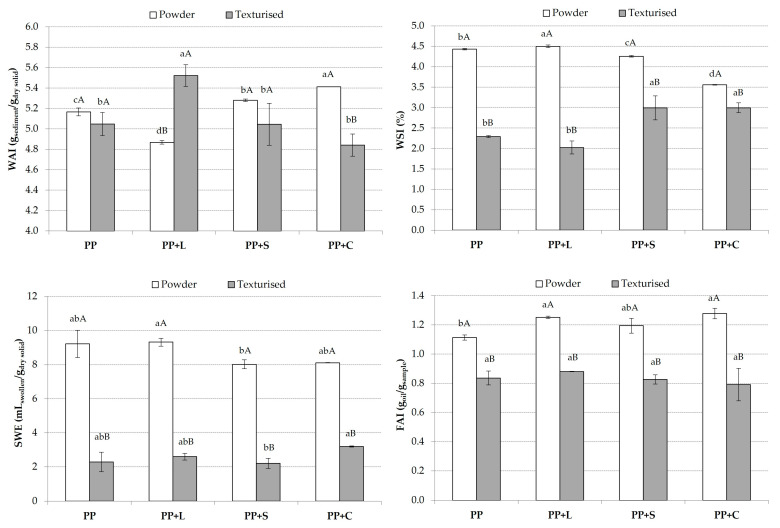
Mean values and standard deviation of water absorption index (WAI), water solubility index (WSI), swelling index (SWE) and fat absorption index (FAI) of powder and texturised protein. For each parameter, the same small letters indicate homogeneous groups established by ANOVA (*p* < 0.05) comparing formulations (PP, PP+L, PP+S and PP+C) of powder and texturised protein. For each sample (PP, PP+L, PP+S and PP+C) and parameter, the same capitals letter indicate homogeneous groups established by ANOVA (*p* < 0.05) comparing powder and texturised protein. PP, pea protein; PP+L, pea protein with lucerne; PP+S, pea protein with spinach; PP+C, pea protein with chlorella.

**Figure 5 foods-12-01303-f005:**
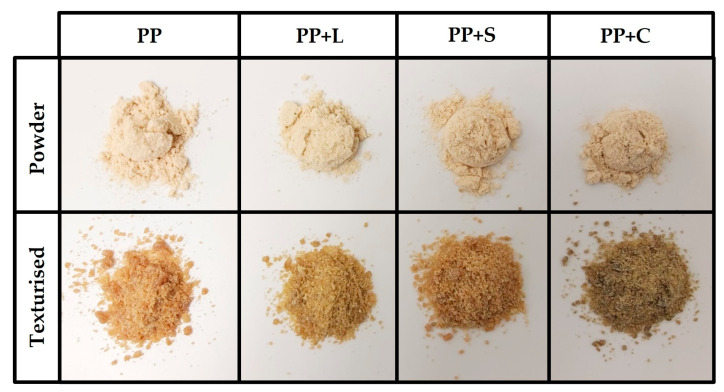
Powder and texturised samples. PP, pea protein; PP+L, pea protein with lucerne; PP+S, pea protein with spinach; PP+C, pea protein with chlorella.

**Figure 6 foods-12-01303-f006:**
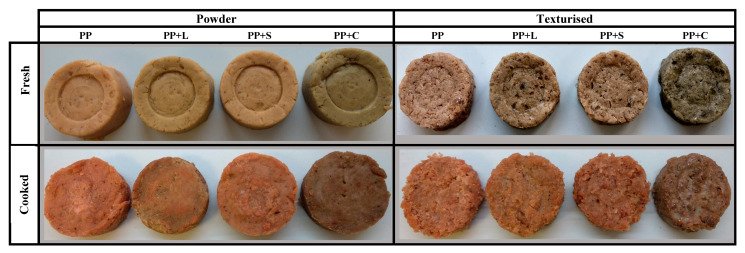
Appearance of fresh and cooked vegetal hamburgers with powder and texturised protein. PP: pea protein, PP+L: pea protein with lucerne, PP+S: pea protein with spinach, PP+C: pea protein with Chlorella.

**Figure 7 foods-12-01303-f007:**
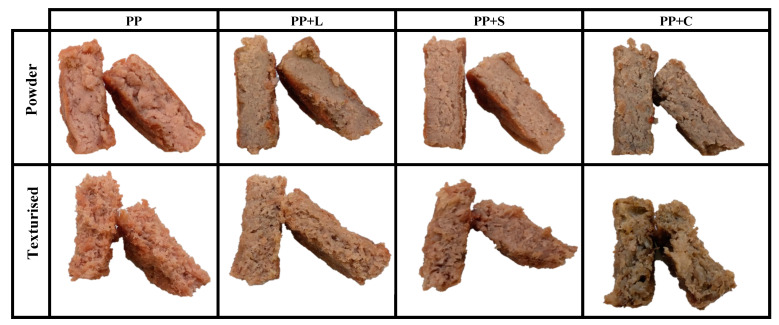
Fibrousness of cooked vegetal hamburgers with powder and texturised protein. PP: pea protein, PP+L: pea protein with lucerne, PP+S: pea protein with spinach, PP+C: pea protein with Chlorella.

**Table 1 foods-12-01303-t001:** Vegetal burger formulation (*w*:*w*, %).

		Protein	Methylcellulose ^1^	Spice Mixture	Olive Oil	Total
Powder	PP	38	50	2	10	100
	PP+L	38	50	2	10	100
	PP+S	38	50	2	10	100
	PP+C	38	50	2	10	100
Texturised	PP	38	50	2	10	100
	PP+L	38	50	2	10	100
	PP+S	38	50	2	10	100
	PP+C	38	50	2	10	100
Spice mixture						
Salt		0.60				
Pepper		0.19				
Garlic powder		0.60				
Brewer’s yeast		0.60				

PP (pea protein), PP+L (pea protein with 1% lucerne), PP+S (pea protein with 1% spinach) and PP+C (pea protein with 1% Chlorella).^1^ Methylcellulose 3% solution.

**Table 2 foods-12-01303-t002:** Sensory attributes definition.

Attributes	Definition	References
Own colour	Similarity of the colour tone of the sample to the characteristic colour of a pea-based vegetal hamburger. Looking at the colour in the cut.	0—light; 5—light brown (characteristic of this product) 10—dark/greenish
Brightness	Amount of reflected light.	0—dull; 10—very shiny
Fibrousness	Number of fibres perceived in the sample.	0—not fibrous; 10—very fibrous
Overall odour intensity	Overall odour intensity of the sample.	0—not perceptible; 10—very intense
Legume odour	Association with legumes.	
Spiced odour	Odour is associated with the olfactory perception of several spices in the hamburger sample.	
Vegetal odour	Association with herbaceous vegetables.	
Meat odour	Similarities perceived with burger meat.	0—not similar to meat burger; 10—very similar to meat burger
Salty	Taste sensation associated with the presence of sodium chloride in the food.	0—not perceived; 5—normal saltiness; 10—very intense
Sweet	Association with sucrose.	0—not perceived; 10—very intense
Bitter	Association with caffeine.	
Umami	Taste sensation produced by monosodium glutamate. Induces salivation and a velvety sensation on the tongue.	
Overall flavour intensity	Overall flavour intensity of the sample.	
Legume flavour	Association with legumes.	
Spiced flavour	Flavour is associated with the olfactory–gustatory perception of several spices in the hamburger sample.	
Vegetable flavour	Association with herbaceous vegetables.	
Meat flavour	Similarities are perceived with burger meat.	0—not similar to meat burger; 10—very similar to meat burger
Hardness	Force required to deform or compress a substance between the teeth.	0—easily compressible; 5—characteristic of minced meat; 10—not compressible
Juiciness	Associated with the amount of water and/or fat contained in a food.	0—completely dry; 10—very moist
Chewiness	Time required to reduce the size of food until it is swallowed.	0—no chewing required for swallowing; 10—requires significant chewing for swallowing
Gumminess	Effort to swallow a soft product.	0—little; 10—very much
Adhesiveness	Work required by the tongue to dislodge a product stuck on the palate or teeth.	
Graininess	Quantity of particles released after chewing 5 times.	0—no particles are perceived; 10—very grainy, mealy mouthfeel

**Table 3 foods-12-01303-t003:** Mean values (and standard deviations) of barrel temperatures (T_1_, T_2_), melt pressure (P) and specific mechanical energy (SME) of the studied samples.

Sample	T_1_ (°C)	T_2_ (°C)	P (Pa)	SME (J/g)
PP	120.2 (0.4) ^a^	121.4 (0.5) ^a^	27 (5) ^a^	1061 (9) ^d^
PP+L	120.3 (0.3) ^a^	122.0 (0.3) ^a^	24 (7) ^a^	1906 (10) ^c^
PP+S	120.7 (0.2) ^a^	122.8 (0.7) ^a^	29 (9) ^a^	2244 (16) ^a^
PP+C	120.5 (0.5) ^a^	122.0 (0.4) ^a^	26 (7) ^a^	1996 (8) ^b^

The same letter in superscript within the column indicates homogeneous groups established by ANOVA (*p* < 0.05). PP, pea protein; PP+L, pea protein with lucerne; PP+S, pea protein with spinach; PP+C, pea protein with chlorella.

**Table 4 foods-12-01303-t004:** Mean values (and standard deviations) of moisture (x_w_, g_water_/100g_sample_) and hygroscopicity after 1, 4 and 7 d (Hy_1d_, Hy_4d_ and Hy_7d_, gwater/100 gdry solid) of powder and texturised protein with different formulations (PP, PP+L, PP+S and PP+C). PP, pea protein; PP+L, pea protein with lucerne; PP+S, pea protein with spinach; PP+C, pea protein with chlorella.

	Sample	x_w_	Hy_1d_	Hy_4d_	Hy_7d_
Powder	pp	5.80 (0.08) ^abB^	15.4 (0.5) ^aA^	23.73 (0.14) ^abA^	26.2 (0.7) ^aA^
	PP+L	5.6 (0.2) ^abB^	15.7 (0.2) ^aA^	24.43(0.17) ^aA^	26.22 (0.12) ^aA^
	PP+S	6.1 (0.5) ^aB^	15.52 (0.12) ^aA^	23.8 (0.5) ^abA^	26.72 (0.12) ^aA^
	PP+C	5.31 (0.08) ^bB^	15.0 (0.4) ^aA^	23.16 (0.02) ^bA^	26.33 (0.04) ^aA^
Texturised	pp	10.4 (0.8) ^abA^	8.482 (0.009) ^abB^	16.28 (0.18) ^bB^	18.92 (0.16) ^aB^
	PP+L	9.4 (0.4) ^bA^	8.7 (0.2) ^aB^	17.3 (0.2) ^aB^	19.5 (0.5) ^aB^
	PP+S	10.61 (0.13) ^abA^	7.90 (0.07) ^cB^	16.58 (0.05) ^bB^	19.309 (0.006) ^aB^
	PP+C	11.3 (0.9) ^aA^	8.1 (0.2) ^bcB^	16.3 (0.4) ^bB^	18.96 (0.12) ^aB^

For each parameter, the same small letters indicate homogeneous groups established by ANOVA (*p* < 0.05) comparing formulations (PP, PP+L, PP+S and PP+C) of powder and texturised protein. For each sample (PP, PP+L, PP+S and PP+C) and parameter, the same capital letters indicate homogeneous groups established by ANOVA (*p* < 0.05) comparing powder and texturised protein.

**Table 5 foods-12-01303-t005:** Mean values (and standard deviations) of colour coordinates (L*, a* and b*) and total colour differences (ΔE_1_ and ΔE_2_) of powder and texturised protein samples with different formulations (PP, PP+L, PP+S and PP+C). PP, pea protein; PP+L, pea protein with lucerne; PP+S, pea protein with spinach; PP+C, pea protein with chlorella.

	Sample	L*	a*	b*	C*	h*	ΔE_1_	ΔE_2_
Powder	pp	79.42 (0.02) ^aA^	4.51 (0.02) ^aB^	21.27 (0.04) ^bB^	21.74 (0.04) ^bB^	78.03 (0.07) ^dA^		
	PP+L	77.57 (0.05) ^cA^	3.21 (0.02) ^cB^	21.75 (0.07) ^aB^	21.99 (0.07) ^aB^	81.61 (0.02) ^bA^		2.28 (0.04) ^bB^
	PP+S	78.68 (0.04) ^bA^	3.81 (0.06) ^bB^	20.66 (0.07) ^cB^	21.01 (0.08) ^cB^	79.54 (0.14) ^cA^		1.19 (0.05) ^cB^
	PP+C	76.19 (0.02) ^dA^	2.71 (0.03) ^dA^	19.10 (0.04) ^dB^	19.29 (0.03) ^dB^	81.93 (0.09) ^aB^		4.288 (0.012) ^aB^
Texturised	pp	55.9 (0.2) ^bB^	10.71 (0.07) ^aA^	27.0 (0.5) ^bA^	29.1 (0.5) ^aA^	68.4 (0.2) ^dB^	25.01 (0.08) ^a^	
	PP+L	57.8 (0.2) ^aB^	5.58 (0.06) ^cA^	28.5 (0.05) ^aA^	29.04 (0.06) ^aA^	78.93 (0.12) ^bB^	21.0 (0.2) ^c^	5.55 (0.12) ^bA^
	PP+S	58.2 (0.2) ^aB^	7.12 (0.12) ^bA^	26.85 (0.02) ^bA^	27.78 (0.02) ^bA^	75.1 (0.2) ^cB^	21.7 (0.2) ^b^	4.3 (0.2) ^cA^
	PP+C	54.8 (0.4) ^cB^	1.51 (0.02) ^dB^	21.2 (0.2) ^cA^	21.2 (0.2) ^cA^	85.93 (0.06) ^aA^	21.5 (0.4) ^b^	11.3 (0.2) ^aA^

L*: lightening, a*: red-green; b*: yellow-blue, C*: chroma, h*: °hue. For each parameter, the same small letters indicate homogeneous groups established by ANOVA (*p* < 0.05), comparing formulations (PP, PP+L, PP+S and PP+C) of powder and texturised protein. For each sample (PP, PP+L, PP+S and PP+C) and parameter, the same capital letter indicates homogeneous groups established by ANOVA (*p* < 0.05), comparing powder and texturised protein.

**Table 7 foods-12-01303-t007:** Mean values (and standard deviations) of physical properties for cooked vegetal hamburger samples manufactured using enriched pea protein (PP, PP+L, PP+S and PP+C) in powder and texturised forms. PP, pea protein; PP+L, pea protein with lucerne; PP+S, pea protein with spinach; PP+C, pea protein with Chlorella.

	Powder				Texturised			
	PP	PP+L	PP+S	PP+C	PP	PP+L	PP+S	PP+C
L*	50.46 (1.34) ^aA^	47.60 (1.94) ^bA^	50.44 (1.52) ^aA^	45.62 (1.78) ^bA^	48.02 (1.89) ^aB^	48.08 (0.96) ^aA^	46.98 (1.82) ^aB^	44.16 (1.56) ^bA^
a*	9.68 (0.95) ^aA^	5.78 (0.87) ^cA^	7.86 (1.18) ^bA^	3.73 (0.53) ^dA^	8.90 (0.37) ^aB^	5.73 (0.56) ^cA^	6.91 (0.58) ^bA^	2.98 (0.42) ^dB^
b*	13.80 (0.65) ^aA^	13.65 (0.47) ^aA^	14.56 (0.98) ^aA^	8.08 (0.44) ^bA^	13.15 (1.00) ^aA^	12.80 (0.89) ^aB^	10.76 (0.50) ^bB^	7.24 (0.38) ^cB^
C*	16.89 (0.55) ^aA^	14.85 (0.29) ^bA^	16.55 (1.35) ^aA^	8.91 (0.56) ^cA^	15.90 (0.80) ^aB^	14.04 (0.76) ^bB^	12.79 (0.65) ^cB^	7.83 (0.45) ^dB^
h*	55.01 (3.42) ^cA^	67.09 (3.65) ^aA^	61.80 (2.50) ^bA^	65.31 (2.57) ^ab^	55.85 (2.56) ^bA^	65.86 (2.96) ^aA^	57.34 (1.81) ^bB^	67.74 (2.48) ^aA^
CL	14.63 (1.68) ^aB^	12.46 (1.73) ^abB^	10.37 (1.98) ^bB^	11.80 (1.66) ^bB^	27.61 (1.07) ^abA^	24.88 (1.34) ^cA^	26.20 (0.72) ^bcA^	28.13 (1.13) ^aA^
Mechanical properties								
HA1	1000.13 (140.47) ^aA^	1033.56 (195.79) ^aA^	1012.19 (147.56) ^aA^	961.19 (156.13) ^aA^	536.44 (101.97) ^aB^	502.19 (91.62) ^aB^	598.75 (169.56) ^aB^	594.13 (78.64) ^aB^
HA2	903.75 (176.55) ^aA^	912.88 (170.18) ^aA^	896.50 (133.04) ^aA^	851.75 (139.70) ^aA^	472.06 (94.00) ^aB^	435.81 (82.50) ^aB^	519.13 (151.75) ^aB^	514.25 (80.71) ^aB^
AD	0.17 (0.15) ^aA^	0.13 (0.11) ^aA^	0.21 (0.12) ^aA^	0.17 (0.14) ^aA^	0.15 (0.11) ^aA^	0.07 (0.05) ^aA^	0.17 (0.09) ^aA^	0.16 (0.09) ^aA^
RE	0.36 (0.07) ^aA^	0.40 (0.04) ^aA^	0.38 (0.03) ^aA^	0.40 (0.04) ^aA^	0.28 (0.04) ^aB^	0.19 (0.02) ^bB^	0.22 (0.01) ^bB^	0.21 (0.02) ^bB^
CO	0.68 (0.14) ^aA^	0.74 (0.03) ^aA^	0.74 (0.03) ^aA^	0.73 (0.05) ^aA^	0.60 (0.04) ^aA^	0.54 (0.03) ^bB^	0.57 (0.04) ^abB^	0.53 (0.04) ^bB^
EL	4.42 (0.21) ^aA^	4.52 (0.17) ^aA^	4.54 (0.07) ^aA^	4.57 (0.09) ^aA^	3.79 (0.43) ^aB^	3.52 (0.77) ^aB^	3.61 (0.37) ^aB^	3.58 (0.20) ^aB^
GU	658.06 (175.66) ^aA^	750.81 (139.40) ^aA^	729.44 (114.87) ^aA^	695.44 (139.64) ^aA^	325.25 (70.57) ^aB^	271.75 (59.14) ^aB^	355.19 (98.93) ^aB^	323.50 (64.95) ^aB^
CH	25.39 (2.15) ^cA^	35.81 (2.89) ^aA^	35.61 (3.65) ^aA^	30.71 (3.75) ^bA^	11.60 (1.42) ^aB^	9.33 (0.98) ^bB^	11.69 (0.59) ^aB^	10.79 (0.99) ^aB^

L*: lightening, a*: red-green; b*: yellow-blue, C*: chroma, h*: °hue, CL: cooking loss. Mechanical properties: HA1: first-bite hardness 1, HA2: second-bite hardness, AD: adhesiveness, RE: resilience, CO: cohesiveness, EL: elasticity, GU: gumminess, CH: chewiness. For each parameter, the same small letters indicate homogeneous groups established by ANOVA (*p* < 0.05) comparing formulations (PP, PP+L, PP+S and PP+C) of powder and texturised protein. For each sample (PP, PP+L, PP+S and PP+C) and parameter, the same capital letter indicates homogeneous groups established by ANOVA (*p* < 0.05) comparing powder and texturised samples.

**Table 8 foods-12-01303-t008:** Sensory profile of cooked vegetal hamburgers made with enriched pea protein. Mean values and SEM are indicated.

	Protein Format (PF)			Formulation (FM)					*p*-Value		
	Powder	Texturised	SEM	PP	PP+L	PP+S	PP+C	SEM	PF	FM	PF × FM
Own colour	6.92	7.60	0.018	5.59 ^a^	7.61 ^c^	6.82 ^b^	9.02 ^d^	0.025	0.000	0.000	0.000
Brightness	3.41	4.25	0.023	3.58 ^a^	3.64 ^a^	3.64 ^a^	4.46 ^b^	0.032	0.000	0.000	0.001
Fibrousness	0.00	4.11	0.024	2.05	2.05	2.05	2.05	0.033	0.000	1.000	1.000
Overall odour	8.03	8.25	0.024	7.93 ^b^	7.76 ^a^	8.11 ^bc^	8.75 ^d^	0.034	0.000	0.000	0.000
Legume odour	3.85	1.02	0.026	2.98 ^c^	2.19 ^b^	2.89 ^c^	1.66 ^a^	0.037	0.000	0.000	0.000
Spiced odour	1.71	2.13	0.018	2.13 ^c^	1.86 ^b^	2.13 ^c^	1.58 ^a^	0.026	0.000	0.000	0.000
Vegetal odour	0.65	1.18	0.021	0.01 ^a^	0.68 ^c^	0.48 ^b^	2.48 ^d^	0.030	0.000	0.000	0.000
Meat odour	0.80	2.73	0.032	1.53 ^a^	1.56 ^a^	1.75 ^b^	2.22 ^c^	0.045	0.000	0.000	0.000
Salty	4.54	5.02	0.031	4.73 ^b^	4.93 ^c^	4.95 ^c^	4.50 ^a^	0.044	0.000	0.000	0.000
Sweet	1.68	0.71	0.024	1.38 ^c^	1.20 ^b^	0.90 ^a^	1.29 ^bc^	0.034	0.000	0.000	0.000
Bitter	1.10	0.76	0.024	0.75 ^a^	1.05 ^c^	0.93 ^b^	0.98 ^bc^	0.033	0.000	0.000	0.000
Umami	1.12	1.87	0.024	1.56 ^b^	1.53 ^b^	1.55 ^b^	1.32 ^a^	0.033	0.000	0.000	0.000
Overall flavour	6.50	6.79	0.024	6.46 ^a^	6.42 ^a^	6.68 ^b^	7.01 ^c^	0.034	0.000	0.000	0.000
Legume flavour	4.57	0.63	0.020	3.18 ^c^	2.65 ^b^	2.62 ^b^	1.90 ^a^	0.029	0.000	0.000	0.013
Spiced flavour	1.71	2.08	0.034	2.05 ^b^	2.02 ^b^	1.95 ^b^	1.58 ^a^	0.048	0.000	0.000	0.000
Vegetable flavour	0.89	1.82	0.023	0.09 ^a^	1.38 ^c^	0.93 ^b^	3.02 ^d^	0.032	0.000	0.000	0.000
Meat flavour	0.00	2.59	0.022	1.15 ^a^	1.58 ^b^	1.33 ^c^	1.12 ^a^	0.031	0.000	0.000	0.000
Hardness	3.40	5.68	0.033	4.08 ^a^	4.65 ^b^	4.66 ^b^	4.75 ^b^	0.046	0.000	0.000	0.958
Juiciness	1.10	4.94	0.041	2.67 ^a^	3.00 ^b^	3.12 ^bc^	3.29 ^c^	0.058	0.000	0.000	0.000
Chewiness	2.29	5.73	0.028	3.52 ^a^	4.21 ^bc^	4.08 ^b^	4.24 ^c^	0.040	0.000	0.000	0.000
Gumminess	0.00	0.17	0.018	0.17 ^b^	0.00 ^a^	0.16 ^b^	0.00 ^a^	0.025	0.000	0.000	0.000
Adhesiveness	3.28	0.00	0.017	2.01 ^d^	1.73 ^c^	1.59 ^b^	1.23 ^a^	0.024	0.000	0.000	0.000
Graininess	7.86	3.09	0.024	5.48 ^b^	6.26 ^c^	5.40 ^b^	4.75 ^a^	0.034	0.000	0.000	0.000

Formulations: PP (pea protein), PP+L (pea protein with 1% lucerne), PP+S (pea protein with 1% spinach) and PP+C (pea protein with 1% Chlorella). Values within a formulation row with different superscripts significantly differ at *p* < 0.05. a,b,c,d: Formulation effect.

**Table 9 foods-12-01303-t009:** Pearson correlation coefficients between parameters of proteins and parameters of vegetal hamburgers. WHC: water-holding capacity, CL: cooking loss, HA1: first bite hardness 1, HA2: second bite hardness, AD: adhesiveness, RE: resilience, CO: cohesiveness, EL: elasticity, GU: gumminess, CH: chewiness.

	Protein				
Vegetal Hamburgers	x_w_	WAI	WSI	SWE	FAI
CL	0.9557 *	−0.2575	−0.8277 *	−0.9334 *	−0.9545 *
WHC	−0.9559 *	0.3530	0.7954 *	0.9180 *	0.9356 *
HA1	−0.8434 *	−0.0048	0.8804 *	0.9240 *	0.8547 *
HA2	−0.8414 *	0.0004	0.8755 *	0.9249 *	0.8479 *
AD	−0.1265	−0.2122	0.3984	0.2354	0.1463
RE	−0.8677 *	0.0436	0.7878 *	0.8760 *	0.8829 *
CO	−0.8182 *	0.0799	0.7285 *	0.8231 *	0.8668 *
EL	−0.8595 *	0.0682	0.7808 *	0.8673 *	0.8729 *
GU	−0.8690 *	0.0017	0.8808 *	0.9290 *	0.8945 *
CH	−0.8998 *	0.0328	0.8752 *	0.9271 *	0.9377 *

* Correlation is significant at the 0.05 level.

## Data Availability

The data presented in this study are available upon request from the corresponding author.

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
