# Peer review of "Enriched Pea Protein Texturing: Physicochemical Characteristics and Application as a Substitute for Meat in Hamburgers"

_foods, 2023, doi:10.3390/foods12061303_

Round 1
Reviewer 1 Report
General comments:
The aim of this article is to evaluate the effect of pea protein (PP) enriched with lucerne (L), spinach (S) and Chlorella (C), in powder and texturised form, on the physicochemical properties and extrusion parameters and to evaluate its technological and sensory quality as a meat substitute in hamburgers. It is commendable that this article has a clear structure and displays rich data and charts. However, this article also has some problems, and requires a revision.
Specific comments:
1. The introduction section needs to be reconsidered to ensure its logic and necessity.
2. Materials and methods section 2.3 and 2.5, each test would be better listed separately.
3. Has the optimization of extrusion conditions been done? What are the optimal extrusion conditions?
4. Why were lucerne (L), spinach (S) and Chlorella (C) added at 1% chosen for the study?
5. How are the conditions for making hamburgers determined?
6. Please note the format of the writing and check again carefully before the article is officially published. For example, Page 3, Line 135、Page 4, Line 168、Page 5, 225.
7. Please rewrite Page 5, Line 187-190.
8. Please rewrite the equation in Page 5, Line 204.
Author Response
Please find notes in the annex.

Reviewer 2 Report
I am very grateful you for the invitation to review the manuscript foods-2254770 by Peñaranda and coauthors "Enriched pea protein texturing: physicochemical characteristics and application as a substitute for meat in hamburgers”. The aim of this study was to evaluate the effect of pea protein enriched with lucerne, spinach and Chlorella, in powder and texturised form, on the physicochemical properties and extrusion parameters and to evaluate its technological and sensory quality as a meat substitute in hamburgers. The work is interesting but needs adjustments to increase the quality of the material.
Comments:
- Abstract: Please include the nutritional aspects related to the consumption of vegetable proteins.
- Abstract: detail better the “environmental and animal concerns”.
- Line 15: Check the term “similar” and consider “analogues” or another appropriate and standardized term throughout the work.
- Line 18: “enrich the formulation” in what sense??
- Lines 18-21: Protein effect on extrusion parameter? The process parameters are usually evaluated in the characteristics of the products.
- Abstract: Please indicate in the abstract a brief step-by-step about the work.
- Lines 22-23: Indicate whether this refers to the patty or extruded material.
- Line 23: Textured samples refers to the hamburger or extruded material?
- Abstract: Include a brief conclusion to this item.
- Lines 30-31: Change the repeated keywords by different words from the title.
- Lines 38-41: Indicate the conversion of protein to meat. Open sentences without scientific data should be avoided.
- Lines 34-43: More important than highlighting the negative aspects of the livestock sector (which despite the points presented feeds a large portion of the population), it is important to highlight the vegetable protein consumption market, aspects related to its consumption, growth, etc.
- Introduction: It is important to highlight the role of proteins in the elaboration of meat analogs.
- Lines 51-53: specify the increase.
- Line 118: Remove the letter “w” at the end of the sentence.
- Line 237: Standardize the name of the hamburger throughout the text (the use of the following terms is verified: fortified veggie burger preparations; meat substitutes; plant-based, analogues, plant-based burgers, etc.). Check the most suitable.
- Line 240: Standardize the unit of time throughout the article (min/minutes, and others).
- Line 310: Please standardize the use of functional and technological nomenclatures.
- Line 358-359: The content of the sentences is not clear: “thus reducing the molecular degradation in samples”.
- Lines 373-376: The phenomenon involved in the reduction is not clear. Explain better, please.
- Lines 378-389: It is not clear the role of the components and the influence on the color. Detail the aspects better.
- Line 485: insert figure after the citation.
- Line 500: Including Maillard?
- Standardize sensory and organoleptic throughout the article.
Author Response
Please find notes in the annex.

Round 2
Reviewer 1 Report
The manuscript can be published now.